# Anti-Cancer Auto-Antibodies: Roles, Applications and Open Issues

**DOI:** 10.3390/cancers13040813

**Published:** 2021-02-15

**Authors:** Hugo de Jonge, Luisa Iamele, Maristella Maggi, Greta Pessino, Claudia Scotti

**Affiliations:** Unit of Immunology and General Pathology, Department of Molecular Medicine, University of Pavia, 27100 Pavia, Italy; hugo.dejonge@unipv.it (H.d.J.); luisa.iamele@unipv.it (L.I.); maristella.maggi@unipv.it (M.M.); greta.pessino01@universitadipavia.it (G.P.)

**Keywords:** auto-antibodies, cancer, tumor-associated antigens, tumor-specific antigens

## Abstract

**Simple Summary:**

Cancer is one of the main causes of death worldwide and early detection is crucial for effective treatment. Scientists have therefore focused on the identification of circulating cancer-specific molecules, so-called markers, that can be detected with non- or less-invasive techniques. One attractive emerging marker is represented by circulating antibodies against molecules that are specific for the tumor cells and not produced by normal cells. Many such auto-antibodies have been discovered, but still not much is known about their significance and role in either promoting or inhibiting tumor growth. The aim of this review is to summarize the current knowledge on auto-antibodies in different cancer types and on their current or possible utilization for effective cancer management. Reference to autoimmune diseases will also be made, as they share with cancer the presence of such auto-antibodies.

**Abstract:**

Auto-antibodies are classically associated with autoimmune diseases, where they are an integral part of diagnostic panels. However, recent evidence is accumulating on the presence of auto-antibodies against single or selected panels of auto-antigens in many types of cancer. Auto-antibodies might initially represent an epiphenomenon derived from the inflammatory environment induced by the tumor. However, their effect on tumor evolution can be crucial, as is discussed in this paper. It has been demonstrated that some of these auto-antibodies can be used for early detection and cancer staging, as well as for monitoring of cancer regression during treatment and follow up. Interestingly, certain auto-antibodies were found to promote cancer progression and metastasis, while others contribute to the body’s defense against it. Moreover, auto-antibodies are of a polyclonal nature, which means that often several antibodies are involved in the response to a single tumor antigen. Dissection of these antibody specificities is now possible, allowing their identification at the genetic, structural, and epitope levels. In this review, we report the evidence available on the presence of auto-antibodies in the main cancer types and discuss some of the open issues that still need to be addressed by the research community.

## 1. Introduction

Cancer is one of the leading causes of death worldwide along with cardiovascular diseases. Huge progress has been made during the last few decades towards early detection, effective treatment, and follow-up, giving oncologists powerful tools in the fight against cancer. However, more research is needed to reduce mortality and increase quality of life and disease-free survival for patients. In this perspective, the search for new markers as indicators for the presence of tumors and as tools to monitor disease progression is essential. Moreover, the discovery of each new marker opens the possibility to shed light on potentially new pathophysiological mechanisms. This is particularly true when antibodies are considered, because of the central role they have in the immune response. Physiologically, they represent a key defense mechanism against infectious diseases, recognizing molecules of exogenous microorganisms as non-self. Their effect is instead detrimental when they target the host tissues in autoimmune diseases. In these pathological conditions, auto-antibodies cause inflammation in joints, such as in rheumatoid arthritis (RA), or can affect the lungs, blood cells, nerves, and kidneys in systemic lupus erythematosus (SLE), or intestines, such as in inflammatory bowel disease (IBD). Worth noting is that patients affected by these diseases have a significantly modified risk to develop cancer [1]. This risk is often increased in SLE, RA, Sjögren syndrome, IBD, and systemic sclerosis, resulting in frequent occurrence of several types of cancers, as reviewed in [2], but is sometimes decreased, such as for breast cancer in SLE, RA, and psoriatic arthritis (PA) [3,4]. It is possible that pre-existing auto-antibodies could directly contribute to promote or suppress cancer progression, but they could also represent the indirect beacons of underlying immunological phenomena closely conditioning cancer development.

The specific pathogenic role of auto-antigens and auto-antibodies is still quite unclear in the realm of autoimmune disorders. Even more intriguing is the fact that auto-antigens and auto-antibodies are being detected in an increasing number of oncological conditions in the absence of overt autoimmune diseases and involving antigens totally unrelated to classical autoimmune conditions. This has raised questions about their role in the progression of cancer and about their potential applications in diagnosis, therapy, and prognosis, as well as about the connection between cancer and autoimmune diseases.

Tumors consist of a complex mixture of both germline-encoded and novel somatically generated antigens. The first class typically derive from proteins that are not antigenic in normal cells. However, when tumor cells start expressing them well above normal levels, or in places where they become exposed to the immune system, while normally they are not, they can become so called tumor-associated antigens (TAAs). The second class of antigens are derived from normal genes by somatic mutation, deletion, or epigenetic modifications and are called tumor-specific antigens (TSAs). On one side, it is not surprising that newly generated cancer antigens can induce an immune response; being novel sequences, they can be recognized as non-self by the immune system, as if they were exogenous molecules. It is more challenging to explain how unmodified cancer TAAs can induce an immune response and how both these phenomena occur only in subgroups of patients. In this respect, the current views concerning the pathogenic mechanisms triggering autoimmune diseases can be useful. Like cancer, even autoimmune diseases are derived from a combination of environmental and genetic factors. Polymorphisms in various genes can result in defective regulation or reduced threshold for lymphocyte activation, and environmental factors (e.g., infections, traumas) initiate or augment activation of self-reactive lymphocytes that have escaped control and that can react against auto-antigens [5]. A role for extracellular vesicles in the process of auto-antibody production has also been proposed [6].

Independently from the mechanism, the capacity of the immune system to detect TSAs and TAAs enables immunosurveillance, the process whereby the body can normally remove newly formed tumor cells well before their growth becomes uncontrolled. The escape from immunosurveillance is, in fact, one of the major mechanisms leading to tumor growth and, in recent years, has been tackled by some innovative immunotherapeutic approaches, among which are anti-PD-1 antibodies [7].

Some tumor antigens are expressed in such a high percentage of tumors that they are called “universal antigens”. Examples of antigens expressed in more than 50% of tumor types and able to induce an immune response are p53, NY-ESO-1, survivin, and MART-1 [8,9,10,11].

The concept of the host immune system generating anti-cancer immune cells and antibodies [12] is one of the pillars of immune-based targeted therapies [13,14], as cell- and antibody-based drugs have been shown to inhibit cancer growth through different complementary mechanisms. However, there is growing evidence suggesting that B cells and antibodies can also be involved in tumor promotion and resistance to cancer therapy [15,16,17], with the observation that B cell depletion can suppress tumor growth in mice [18]. In ovarian cancer, the presence of CD20^+^ B cells has been associated with an increased survival, while in contrast, the presence of regulatory B cells (B_regs_) induces immunosuppressive effects, supporting tumor growth [19]. The B-cell response to cancer antigens, therefore, looks to have complex functional features, likely dependent on the specific antigen, or antigens, involved. One useful consequence of the engagement of B cells in the anti-tumor response is that, independently from their positive or negative role, new anti-tumor-specific antibodies are typically generated in the host organism. In cases where they can be detected in the bloodstream, they can herald tumor presence, facilitating early detection or aiding in the identification of an ongoing change within the tumor, for example, towards a metastatic phenotype. Treatment can also influence anti-cancer auto-antibody levels, as demonstrated by Evans et al. [20], where the treatment modality was shown to have a different effect.

In this review, we present current knowledge on anti-cancer antibodies in different tumor types, focusing on their potential to act as early beacons of cancer presence. The main tumor antigens known to trigger an immune response in the host and the available observations regarding their role in tumor progression and in metastasis are also analyzed. Finally, a discussion of the most relevant open issues in this field is offered.

## 2. Cancer Auto-Antibodies

Many different cancer types have been reported to induce the production of auto-antibodies. In the following paragraphs, a survey of the relevant literature regarding the most common cancer types is proposed in anatomical order from head to foot.

### 2.1. Head and Neck Cancer

Head and neck cancers (HNCs) are neoplasms that most commonly originate from squamous cells that make up the epidermis that line the mucosal surfaces inside the head and neck such as the oral cavity, the larynx and pharynx, as well as the nasal and paranasal spaces. Since these tissues are exposed to external factors, the main risk factors have been identified to be tobacco smoke, alcohol, and environmental pollution. In addition, pathogens such as human papillomavirus (HPA) are often causative for this type of cancer as well [21]. The main treatment of HNC remains one that is based on traditional approaches such as surgery, radiation therapy, and chemotherapy, often combined. However, in the last decade, several specific genetic targets have been identified that allow for targeted immunotherapeutic approaches [22]. This search for specific biomarkers has led to the identification of circulating antibodies specifically recognizing the cancer-testis antigen SP17, which was found to be present in 31% of serum samples taken from patients and not present in serum from the healthy control group [23]. The authors compared it to previously reported anti-p53 auto-antibodies, which were found in only 25% of HNC patients [24]. Most importantly, the SP17 antibodies were detected in patients with regionally confined as well as metastasized disease, making it a promising candidate for diagnostic application, especially when combined with additional biomarkers such as the “classical” cytokines and chemokines (IFN-γ, IL-13, MIP-1β, IP-10) [25].

### 2.2. Central Nervous System Cancers

Central nervous system (CNS) tumors are not easily diagnosed because of the reduced symptomatic manifestations. Moreover, the procedure to obtain a biopsy of suspect masses is a cumbersome, risky, and sometimes impossible procedure. Glioma and meningioma are the most common forms of CNS tumors [26]. O-6-methylguanine-DNA methylase (MGMT) is a biomarker of resistance to chemotherapy in glioma patients. MGMT is involved in DNA repair, and its overexpression and methylation state are routinely investigated in glioma patients to predict drug resistance. A peptide microarray study was conducted to evaluate the production of auto-antibodies against MGMT, and the authors found a good correlation between the presence of the MGMT-2 peptide auto-antibodies and the risk of chemotherapy resistance and disease recurrence [27]. Another study evaluated the proteome signature of meningiomas using human proteome arrays. The study results showed that several proteins overexpressed in grade I and grade II meningiomas are known targets of auto-antibodies, such as IGHG4, CRYM, EFCAB2, STAT6, HDAC7A, and CCNB1 [28]. In a different study, circulating anti-pituitary (APA) and anti-hypothalamus (AHA) antibodies were evaluated in children with CNS tumors. Detectable levels of APA and AHA antibodies were found in patients, but not in healthy controls. In particular, the presence of APA and/or AHA was elevated in patients diagnosed with germinomas, gliomas, and craniopharyngiomas. A similar increase in APA and/or AHA was described in autoimmune pituitary conditions [29].

### 2.3. Gastrointestinal Tumors

Gastrointestinal cancers are among the most frequent cancers, and they are the third leading cause of cancer-related death worldwide, with an estimation of growing incidence in the next few years. Diffuse gastrointestinal cancers are mainly related to abnormal expression or mutations of E-cadherin, a cell–cell adhesion protein. Several studies have analyzed auto-antibodies’ correlation with gastrointestinal cancer. The most frequently tested and detected are anti-p53 auto-antibodies, but their abundance does not always correlate with gastrointestinal cancer prognosis, stage, or grade, with some exceptions [30]. Adding anti-p53 antibodies to conventional markers significantly improved the overall detection rates of esophageal and colorectal cancers [31]. Stage-related auto-antibody abundance was described for the antigen NY-ESO-1, detectable in significant concentrations in late-stage patients for which the persistence of auto-antibodies is related to poor prognosis. Similarly, anti-AEG-1 auto-antibodies were also mostly found in late-stage patients [32]. Good sensitivity but poor correlation with tumor stage was described for anti-CTAG2, anti-DDX53, anti-MAGEC1, anti-MAGEA3, and GPR78 [32]. Specific studies for the different organs are also available, which are illustrated in the next sections.

#### 2.3.1. Gastric Cancer

Anti-RaIA antibodies have been detected in 15% of gastric cancer cases and were found to be a candidate serum marker for gastric cancer when used in combination with CEA and/or CA19-9 [33]. The presence of anti-RalA antibodies, in combination with CEA and/or CA19-9, was also associated with poor survival in patients with gastric cancer. Even anti-p53 antibodies, when combined with CEA and CA 19-9, can be a good diagnostic tool for gastric cancer [33]. Lymphatic node metastases and distant metastases, but not overall survival, were all significantly associated with the presence of p53 auto-antibodies [34]. The patients in the high titer group of anti-p53 antibodies (>100 U/mL) showed a worse survival than those in the other groups. In a case series of 407 gastric cancer patients [35], auto-antibodies against nine antigens were detected (c-Myc, p16, HSPD1, PTEN, p53, NPM1, ENO1, p62, HCC1.4), while in another series, an optimal prediction model with six TAAs (p62, c-Myc, NPM1, 14-3-3ζ, MDM2, and p16) was designed [36]. A different six-antigen panel, including CTAG1B/CTAG2, DDX53, IGF2BP2, P53P53, and MAGEA3, detected 13% of gastric cancer patients [37]. Hoshino et al. determined a good sensitivity and specificity for a panel of six TAAs: p53, heat shock protein 70, HCC-22-5, peroxiredoxin VI, KM-HN-1, and p90 TAA [38]. In a work published by Wang et al. [39], a panel of four markers, namely, ALDH1B1, UQCRC1, CTAG1, and CENPF, and even ALDH1B1 alone, were found to be good early predictors of the presence of gastric cancer. Finally, the frequency of anti-14-3-3ζ, a gastric cancer auto-antigen identified by serological proteome analysis (SERPA) [40], was significantly higher in the selected gastric cancer group than in the control group [41].

#### 2.3.2. Colorectal Cancer

p53 has also been found to be an autoantigen in colorectal cancer and, in a meta-analysis considering 199 antigens, it was found to be an important auto-antibody for diagnostic purposes in combination with those against c-MYC, cyclin B1, p62, Koc, IMP1, and survivin [42]. Even alone, p53 auto-antibodies were able to predict the likelihood to develop colorectal cancer within 6 years [43], although their presence at recurrence was not found to be a relevant prognostic factor [44]. Disease-free individuals and carriers of metastatic colorectal cancer could be distinguished using ERP44 and TALDO1 antigens [45]. In another multi-panel approach [46], auto-antibodies against 17 tumor antigens (p53, RalA, HSP70, Galectin1, KM-HN-1, NY-ESO-1, p90, Sui1, HSP40, Cyclin B1, HCC-22-5, c-myc, PrxVI, VEGF, HCA25a, p62, and Annexin) were evaluated for their association with colorectal cancer. The highest positive rates were found for p53 (20%), RalA (14%), HSP70 (12%), and Galectin1 (11%). Positive rates increased to 56, 62, 66, 71, and 73% using panels of 6, 9, 11, 14, and 17 antibodies, respectively. Moreover, these auto-antibodies showed relatively high positive rates even during stage 0/I disease (55 and 70% with 6 and 17 antibodies, respectively) [46].

#### 2.3.3. Pancreatic Cancer

Preventing lethality due to pancreatic cancer is one of the most challenging goals in oncology today. The work on immunotherapy is underway to find antigens able to detect and treat this aggressive cancer before it becomes clinically evident, a stage that is most of the time too advanced for successful treatment [47].

MUC4 is an antigen that is normally not expressed in the pancreas but often appears in tumor cells and can induce the production of auto-antibodies and reactive T cells, especially against MUC4 forms carrying aberrant glycosylation patterns or mutations, or differing for alternative splicing [48].

Pancreatic neuroendocrine tumors are rare compared to adenocarcinoma and, as the latter, they are often diagnosed at an advanced clinical stage. Chromogranin A is a widely used tumor marker in these cases, but it has significant limitations. Among five tumor antigens identified (PTEN, EP300-interacting inhibitor of differentiation 3 (EID3), EH domain-containing protein 1, galactoside-binding soluble 9, and BRCA1-associated protein), antibody levels against the EID3 antigen were significantly higher in the patient group than in the healthy donor group, and high levels were related with shorter disease-free survival [49].

#### 2.3.4. Liver Cancer

In a prospective multicenter study including 160 patients carrying hepatocarcinoma (HCC) and 74 healthy controls, the diagnostic potential of a panel of six auto-antibodies against Sui1, p62, RalA, p53, NY-ESO-1, and c-Myc antigens was tested [50]. The combination of α-fetoprotein (AFP) levels with auto-antibody titers against 10 antigens increased the sensitivity for detecting stage 1 HCC by 40.00% and stage 2 by 55.00% over the tumor antigens auto-antibodies panel alone or AFP alone [51].

In 2013, the general concept of “driver genes” was introduced to indicate mutations of about 140 genes that can lead or “drive” carcinogenesis following intragenic mutations, acting on 12 pathways and 3 functions: cell fate, cell survival, and genome maintenance [52]. A typical tumor contains two to eight driver mutations, while all the others do not confer growth advantage. In the case of HCC, seven TAAs representing driver genes (PTCH1, GNA11, PAX5, GNAS, MSH2, Survivin, and P53) were discovered to induce auto-antibody production and were implemented into a predictive model of HCC. Moreover, a nomogram was produced that is useful for bedside evaluation [53]. Another predictive panel of auto-antibodies induced by five antigens, namely, HCC1, P16, P53, P90, and Survivin, was found by another group [54]. Using the predictive score for the presence of HCC on the basis of the panel, derived from a cohort comprising 160 HCC patients and 90 control individuals, the researchers found that diagnosis could have been anticipated by 0.75 year in 16 patients out of the 17 analyzed.

Even single antibodies can help in the diagnosis of HCC. For instance, the levels of anti-sperm-associated antigen 9 (SPAG9) antibodies were significantly higher compared with those in patients with hepatitis/cirrhosis or healthy controls. When combined with anti-SPAG9 antibodies and α-fetoprotein (AFP) levels, the diagnostic specificity and sensitivity were further improved [55].

As is clear from several previously mentioned studies, anti-NY-ESO-1 antibodies are observed in a multitude of malignancies. In intrahepatic cholangiocarcinoma, they were detected in 18.4% of patients, a value significantly higher than that of patients with chronic hepatitis B. Serum NY-ESO-1 antibodies were positively correlated with tumor differentiation, lymphatic metastasis, cTNM stage, and abdominal pain. Importantly, there was a higher cumulative survival rate in patients with serum anti-NY-ESO-1 positivity than in those with serum negativity among the patients with stage III or IV. These data suggest that NY-ESO-1 antibodies might be a helpful tumor marker and a prognostic predictor in intrahepatic cholangiocarcinoma [56].

#### 2.3.5. Esophageal Cancer

Esophagus and esophagogastric junction (EGJ) cancers are the sixth leading cause of cancer-related death and the seventh most frequently diagnosed cancers worldwide. Esophageal cancers are histologically distinguished into carcinomas and adenocarcinomas, and their main causes are obesity, chronic reflux, alcohol, and smoking [57]. Presently, there are no tools for early stage diagnosis since esophageal cancer symptomatology is not specific and the onset of cancer-related symptoms is often a consequence of cancer spread and metastasis. The vast majority of patients with esophageal cancers (>50%) are therefore diagnosed when the cancer has already spread, and only a small portion of patients can undergo curative surgical removal of local cancer masses [57]. Biomarkers such as HER2 and PD-L1 are routinely employed to determine disease prognosis and therapeutic strategy [58]. For example, use of HER2 inhibitory monoclonal antibodies has been proven to be effective to some extent in disease management [59]. The current most specific and reliable biomarkers of esophageal cells’ malignant transformation are aberrant expression of p53, reported in 50–60% of patients; moreover, early stage dysplasia is often associated with high or low expression of p53 [58,60]. Other biomarkers include other cell cycle regulators such as CDKN2A and CCND1, transcription factors (e.g., TOP1, TP63, SOX2, MYC), and chromatin remodeling proteins (e.g., KDM6A). Abnormal expression of biomarkers can stimulate the production of auto-antibodies. Serum auto-antibodies against the p53 protein are mostly found in later-stage patients, predicting poor prognosis and high risk of recurrence. Monitoring of auto-antibodies against p53 in post-surgical patients is predictive of residual tumor cells and recurrence [61]. Anti-TOP48 auto-antibodies were found to be significantly higher in patients with esophageal carcinoma, also in early stage disease, than in the healthy population; hence, anti-TOP48 can be useful for early diagnosis and prognosis [62]. Another marker of esophageal cancer that can be detected in the early stage of the disease is p16, a protein over-expressed mainly in human papillomavirus (HPV)-positive esophageal carcinomas. Anti-p16 auto-antibody levels inversely correlate with cancer stage [58]. The cytosolic protein NY-ESO-1 is a germ line-specific antigen, often overexpressed in cancer, and high levels of NY-ESO-1 specifically correlate with esophageal carcinomas. Anti-NY-ESO-1 auto-antibodies are significantly sensitive and specific and therefore they can be considered good early stage biomarkers for esophageal cancers. Moreover, increased levels of anti-NY-ESO-1 are associated with an advanced cancer stage [63].

The diagnostic value of an autoantibody panel formed by p53, NY-ESO-1, MMP-7, Hsp70, PRDX6, and Bmi-1 in esophagogastric junction adenocarcinoma was evaluated in a training cohort and a validation cohort by enzyme-linked immunosorbent assay [64]. Sensitivity and specificity were 59.0% and 90.5% in the training cohort, and 61.4% and 90.0% in the validation cohort, respectively. A panel consisting of three auto-antibodies (HCCR, C-myc, and MDM2) and three miRNAs (miR-21, miR-223, and miR-375) attained great diagnostic value for esophageal squamous cell carcinoma, with a sensitivity of 69% and a specificity of 90%. The optimal panel of six-member markers was able to effectively discriminate the patients with esophageal squamous cell carcinoma (ESCC) from normal individuals, especially for early esophageal squamous cell carcinoma.

While Hsp-70 auto-antibodies are detected mostly in gastric or colon cancers, the highest levels of anti-Hsp70 are found in esophageal carcinomas and are associated with poor prognosis [58]. Increased expression of fascin is related to tumor invasion and metastasis, with high levels of serum anti-fascin antibodies in esophageal cancers related to poor prognosis [65]. MMP-7 is a metalloproteinase over-expressed in cancers with high propensity to invasion and metastasis, and high serum levels of anti-MMP-7 are found in esophageal cancer patients and are connected to staging and invasiveness [66]. In particular, the authors found that increased MMP-7 expression in esophageal cancer patients was positively correlated to TNM stage III-IV (odds ratio (OR) 3.04, 95% confidence interval (CI) 1.43–6.46, *p* = 0.004).

### 2.4. Thyroid Cancer

Differentiated thyroid carcinoma (DTC) is the most frequent form of endocrine tumor with a good survival rate and low recurrence in previously treated patients. DTC treatment often consists of total or partial surgical removal of the thyroid, followed by iodine-ablation [67]. Cancer clearance and recurrence in DTC is usually monitored by measurement of thyroglobulin (Tg) levels; complete ablation of cancer cells after thyroid removal results in an undetectable level of Tg, and therefore an increase in Tg level indicates only partial cancer removal or cancer recurrence [67,68]. An important aspect in the measurement of Tg levels in DTC patients is the presence of auto-antibodies produced against Tg. Anti-Tg antibodies appear in 10% of normal population and in 15–30% of DTC patients [69]. Patients with a high level of anti-Tg at diagnosis have a higher probability of disease recurrence. Moreover, anti-Tg levels can be monitored to evaluate disease relapse risk; in particular, stable or increasing concentrations of anti-Tg during DTC follow-up is significantly related with disease persistence and recurrence. Conversely, a decrease in anti-Tg after surgery is a sign of good prognosis [68,69].

### 2.5. Lung Cancer

The majority of lung cancer (LC) cases are diagnosed at advanced stages, primarily because earlier stages of the disease are either asymptomatic or symptoms may be attributed to other causes such as infection or long-term effects induced by smoking. Auto-antibodies are proving to be a useful tool for early detection. Most studies in this field focus on two classes: either general LC or non-small cell lung cancer (NSCLC).

In a recent review regarding biomarkers in LC [70], auto-antibodies have been reported to be identified in all histological types and stages. A well-validated auto-antibody panel (p53, NY-ESO-1, CAGE, GBU4-5, Annexin 1, and SOX2), known as EarlyCDT-Lung [71], has been studied in different screening cohorts as an approach to monitor high-risk patients. The panel was further enriched by other antigens (Lmyc2 and cytokeratin 20 or alpha-enolase and cytokeratin 20), which increased sensitivity and specificity [72]. A study comparing six (p53, NY-ESO-1, CAGE, GBU4-5, Annexin I, and SOX2) versus seven (p53, NY-ESO-1, CAGE, GBU4-5, SOX2, HuD, and MAGE A4) auto-antibodies was performed in 2012 [73], demonstrating a superior sensitivity with the latter panel. The EarlyCDT-Lung test was further implemented with a stratification in four risk classes by accurate study of the cut-offs, which provided a 25-fold difference in lung cancer probability between the highest and lowest group [74]. Since 2013, the EarlyCDT-Lung test has become a relevant complementary tool to a Computer Tomography CT scan for detection of early LC and, in 2017, a positive auto-antibody test result was found to reflect a significant increased risk for malignancy in lung nodules of 4 to 20 mm in diameter [75]. Patients at intermediate risk of lung cancer and those who are scheduled for CT surveillance alone could benefit from an auto-antibody search [76]. An ongoing clinical trial (NCT01925625) is currently trying to determine if the use of EarlyCDT-Lung test to identify people at high risk of lung cancer, followed by X-ray and computed tomography scanning, can reduce the incidence of patients with late-stage lung cancer (III and IV) or unclassified presentation (U) at diagnosis, compared to standard practice [77]. The recruitment phase is now closed [77]. The specificity of auto-antibodies was found to be very high, between 91% and 93%, while a lower sensitivity, between 36% and 41%, was found in two studies [78,79].

In LC, also TOP2A, ACTR3, RPS6KA5, and PSIP1 can elicit a humoral immune response and consequently have the potential to serve as a serological biomarkers in early stage tumors [80].

A mixed biomarker panel of three immunogobulin A (IgA) auto-antibodies against BCL7A, TRIM33, and MTERF4, and three IgGs against CTAG1A, DDX4, and MAGEC2, were used for diagnosis of early stage LC and were proven to have 73.5% sensitivity at >85% specificity [81]. Moreover, human epididymis secretory protein 4 (HE4) proved to be an auto-antigen useful for LC diagnosis in high-risk groups, distinguishing the affected group from the control group with a 54.76% sensitivity [82]. Using mass spectrometry, the authors identified a panel of 10 proteins bound to the IgG fraction of lung cancer patients, comprising MUC17, CAMSAP2, KIF13B, SMG1, MED14, ALMS1, GCC2, TIMELESS, TNS1, ATP1A4, and HRNR [83]. This study, on the basis of linear epitope analysis, also highlighted the presence of antigens shared by squamous and non-squamous lung cancer [83]. A different seven-antigen panel (p53, PGP9.5, SOX2, GAGE7, GBU4-5, MAGE A1, CAGE) was also tested to aid early diagnosis of lung adenocarcinoma with ground-glass nodules (GGNs) and/or solid nodules in the Chinese population. The sensitivity and specificity of the auto-antibody assay were 48.6% and 92.7%, respectively [84]. Moreover, in NSCLC, SOX2 and p53 are part of a different panel of auto-antibodies (SOX2, GAGE 7, MAGE A1, and p53), found to be increased in a set of NSCLC [85]. Interestingly, a deficiency of natural antibodies against CD25, Mucin 1 (MUC1), and vascular endothelial growth factor receptor 1 (VEGFR1) has been proposed to contribute to high risk of NSCLC. In fact, levels of auto-antibodies against these antigens have been found to be systematically lower in NSCLC patients than control subjects [86]. Impressively, an optimized panel with four biomarkers (CEA, CA125, Annexin A1-Ab, and Alpha enolase-Ab) was established presenting an area under the receiver operator characteristic curve of 0.897, a sensitivity of 86.5%, a specificity of 82.3%, a positive predictive value (PPV) of 88.3%, a negative predictive value (NPV) of 79.7%, and a diagnostic accuracy of 84.8% for the training group [87]. A model including a three auto-antibodies panel (GREM1, HMGB3, and PSIP1) could also effectively identify cancer cases compared to controls [88].

Treatment with PD-1 immune-check point blockade nivolumab in NSCLC was found to induce the production of auto-antibodies. Early detection (within 30 days) of more than one auto-antibody type among anti-nuclear antigens (ANAs), anti-extractable nuclear antigens (ENAs), and anti-smooth cell antigens (ASMAs), in patients treated with nivolumab-based salvage therapy, associated with prolonged progression-free survival [89].

RNA-sequencing (RNA-seq) based expression profiling of cancer testis antigens (CTAs) in NSCLC indicated activation of 35 previously described CTAs and 55 additional CTAs with no or substantially lower expression in somatic tissues [90]. Some of these proteins (CT47A, PAGE3, VCX, MAGEB1, LIN28B, or C12orf54) were proven to induce the production of auto-antibodies, although in a limited number of cases [91].

The anaplastic lymphoma kinase (ALK) protein is encoded be the ALK gene, which is rearranged in 3–7% cases of NSCLC and is recognized by the immune system as a tumor antigen. Pretreatment ALK antibody titers are inversely correlated with stage of disease, amount of circulating tumor cells, and cumulative incidence of relapse in lymphoma [92]. Patients with NSCLC often have antibodies whose epitopes are within the ALK cytoplasmic domain, but outside the tyrosine kinase domain. In one study, strong anti-ALK antibody responses were detected in 17% ALK-positive and in 0% ALK-negative NSCLC patients [93].

### 2.6. Breast Cancer

Since very early lesions of the breast are undetectable by mammography or ultrasound scan, auto-antibodies represent a promising tool to allow for early diagnosis and monitoring of tumor progression from an in situ to an invasive or metastatic phenotype [94].

HSP60 is one of the breast cancer antigens that stimulate auto-antibody production, and in one study was detected in 31% cases of early stage breast cancer and in 32.6% cases of ductal carcinoma in situ (DCIS) compared to only 4.5% detection in healthy control patients [95]. In a cohort of women diagnosed with triple-negative breast cancer, a panel of antigens was detected after panning the patient sera with MDA-MB-231 cell line lysate. Out of the detected proteins, the highest score was for PI3K and p53 [96]. On the basis of similar evidence, researchers proposed an immunodiagnostic model for the prediction of breast cancer versus benign lesions and control [97]. Another breast cancer auto-antigen is mitochondrial nuclear retrograde regulator 1 (MNRR1), a mitochondrial protein that regulates multiple genes that function in cell migration and cancer metastasis and that is more highly expressed in cell lines derived from aggressive tumors [98]. Serum p53 auto-antibodies have also been shown to be associated with aggressiveness of breast cancer [99].

Gu et al. [100] have shown that anti-cancer antibodies produced by B cells can mediate recruitment of cancer cells derived from the primary breast cancer tumor into draining lymph nodes, thus contributing to the dissemination of the disease. In particular, this pro-metastatic effect was mediated by binding of pathogenic IgGs to glycosylated heat shock protein family A member 4 (HSPA4), a candidate tumor antigen of the HSP70 family. This cell-surface receptor can activate a signaling cascade that is involved in the expression of the CXCR4 ligand stromal-derived factor 1α (SDF1α) in lymph node stromal cells. In patients with breast cancer, high tumor expression of HSPA4 and elevated serum levels of anti-HSPA4 IgG were found to be associated with lymph node metastasis and poor prognosis [100].

### 2.7. Adrenocortical Carcinoma

Adrenocortical carcinoma (ACC) is a rare, but aggressive endocrine malignancy, which often has a negative prognosis especially when diagnosed at an advanced stage. Among the determinants of malignant behavior of this tumor, a critical role is played by molecules modulating cell death and resistance to chemotherapeutic agents. One of these molecules is the steroidogenic factor-1 (SF-1) target gene FATE1, encoding for a protein localized at the interface between mitochondria and endoplasmic reticulum, where it regulates Ca^2+^-dependent and mitotane-induced apoptosis in ACC cells by modulating the distance between the two organelles [101]. FATE1 is expressed at high levels in about 40% of adult ACC and its expression is significantly and inversely correlated with patients’ overall survival. Additionally, FATE1 could be relevant also in other tumors as it has been reported that its silencing increased sensitivity of the NCI-H1155 NSLC cell line to paclitaxel and reduced the viability of a variety of other cancer cell lines. Moreover, circulating antibodies directed against this protein were detected in 3 out of 41 (7.3%) and 4 out of 52 (7.7%) patients with hepatocellular carcinoma in two different studies. Patients with adrenocortical tumors had high tumor FATE1 mRNA expression levels and could mount an immune response against FATE1, as shown by the widespread presence of circulating antibodies directed against this cancer-testis antigen. This is associated with high steroidogenic gene expression, immune cell depletion, and a worse prognosis. High steroid production by FATE1-expressing tumors is likely to create an unfavorable environment for immune cell infiltration and local response against this antigen. On the other hand, FATE1 expression in the most aggressive group of ACC could open new perspectives for immunotherapy using vaccination against this and other cancer neoantigens [101].

### 2.8. Ovarian Cancer

The overall five-year survival rate for ovarian cancer is below 30%, as over 70% of patients are diagnosed with stages III or IV disease [102]. However, subjects diagnosed with localized disease have a survival rate of 75–90%. There is therefore a need to identify biomarkers that have utility for ovarian cancer early detection. At present, cancer antigen 125 (CA125) is the most investigated early detection marker for ovarian cancer and, because the protein detection in circulation has limited sensitivity, tumor-associated auto-antibodies may improve on the performance of CA125 alone. A study by Fortner et al. [102] demonstrated that, when circulating CA125 levels are examined in the context of anti-CA125 antibodies, their ability for the early detection of ovarian cancer may be improved, especially among women with higher anti-CA125 antibody levels. In these early detected cases, lower circulating CA125 levels were also observed among women with higher anti-CA125 antibody levels, consistent with the hypothesis that higher antibody levels may mask detection of circulating antigen. Circulating immune complexes (CIC) to CA125 have been identified, and potential interference with conventional assays for CA125 has been demonstrated.

In a screening involving 20 ovarian cancer patients and 17 controls, circulating auto-antibodies towards P53 and MYC were primarily found, followed by other proteins of the P53 and MYC networks both by a recombinant protein-based assay and by an immunoglobulin-bound protein assay [103]. A study reported by Yang et al. [104] also focused on P53 auto-antibodies. In this case, only 20% to 25% of patients with invasive epithelial ovarian cancer showed elevated levels of auto-antibodies against P53. However, it was also demonstrated that P53 auto-antibody levels could complement CA125 at the time of diagnosis and could provide substantial lead time over CA125 in a fraction of cases, making P53 a possible important member of a panel of biomarkers that would substantially improve on the performance of CA125 alone [104].

Katchman et al. [105] published a screening on sera samples from a total of 94 patients with serous ovarian cancer, 30 with a benign disease, and 92 healthy control samples, searching for auto-antibodies using a custom protein microarray. Eleven auto-antibodies that could distinguish serous ovarian cancer from benign disease (with 32% sensitivity) and healthy controls (45% sensitivity) at 98% specificity were found, five of which have been previously identified in serous ovarian cancer (p53 and CTAG2) or related cancers (NUDT11, PVR, and TRIM39) as contributing to cancer progression. Three auto-antibodies, against NUDT11, PVR, and TRIM39, were consistently selective for serous ovarian cancer with individual sensitivities ranging from 14.7% to 32.4% at 96% specificity. On the basis of these data, the authors suggested how using multiple tumor markers in a single multimodal screening might improve the performance characteristics of the tests [105].

A larger screening was performed with 164 serum samples consisting of 50 late-stage, high-grade serous ovarian cancer (HGSOC); 14 early stage HGSOC; 50 benign ovarian cyst; and 50 healthy control samples to identify novel auto-antibodies by ELISA and Western blot [106]. The results showed that TRIM21 achieved the highest sensitivity in the first validation screening of 33% with 100% specificity. Combining TRIM21 with NY-ESO-1, P53, and PAX8 provided 67% and 56% sensitivity at 94% and 98% specificity, respectively. These four markers resulted in 46% sensitivity with 98% specificity in the second validation cohort while TRIM21 achieved the highest individual sensitivity of 36% [106].

A group of samples of similar size was analyzed, taking into account ascitic fluids collected from 153 patients diagnosed with OC (*n* = 69), other cancers (*n* = 34), and noncancerous conditions (*n* = 50) [107]. In this case, the screening pipeline included initial purification of IgGs from ascitic fluids that were afterwards challenged by performing rounds of selection against an open reading frame (ORF) fragments phage display library to identify the cognate antigens. Reactivity against selected putative antigens was further verified by protein microarray and by ELISA using the whole cohort of 153 ascites samples. This approach allowed for the identification of eight antigenic proteins: CREB3, MRPL46, EXOSC10, BCOR, HMGN2, HIP1R, OLFM4, and KIAA1755, with CREB3 showing the highest sensitivity (86.95%) and specificity (98%). Moreover, a strong association between platinum sensitivity and a higher level of antibodies against BCOR, MRPL46, and CREB3, combined as a single signature, was observed [107].

A different approach consisted in focusing on six proteins (CCL18, CXCL1, C1D, TM4SF1, TIZ, and FXR1) already known for their involvement in various types of malignancies [108]. Serological analysis of recombinant complementary DNA (cDNA) expression library (SEREX) technology was simultaneously used to screen for serum auto-antibodies [109]. It was determined that CCL18 and CXCL1 had a high sensitivity for ovarian cancer diagnosis, but the specificity was not satisfactory; however, C1D, TM4SF1, TIZ, and FXR1 had a high specificity compared with a combination of CCL18 and CXCL1. Therefore, it was hypothesized that the combined use of these different types of markers in the diagnosis of ovarian cancer may compensate for their respective disadvantages [108].

In a study by Wilson et al. [110], a small group of patients with early stage, high-grade serous ovarian cancer was analyzed for IgG-, IgA-, and IgM-mediated autoantibody reactivity, and a high content of IgA auto-antibodies in early stage rather than late-stage disease was highlighted, suggesting they may be useful early indicators. Auto-antibodies against two main antigens were identified and validated as potential new biomarkers of early-stage: heat shock transcription factor 1 (HSF1) and coiled-coil domain-containing 155 (CCDC155). However, this finding could only be partially confirmed in individual patients by an ELISA assay [110].

A panel of 11 auto-antibodies (ICAM3, CTAG2, p53, STYXL1, PVR, POMC, NUDT11, TRIM39, UHMK1, KSR1, and NXF3) provided 45% sensitivity at 98% specificity for serous ovarian cancer [111].

Glucose-regulated protein 78 (GRP78) is a chaperone-assisting protein folding and is normally located in the lumen of the endoplasmic reticulum (ER) [112]. In adult tissues, GRP78 is found at low levels, but in various types of cancer cells, including lung and colon adenocarcinomas, neuroblastoma, and ovarian tumors, GRP78 was found to be overexpressed at the cell surface where it could influence signaling pathways, leading to proliferation and invasion. The levels of GRP78 auto-antibodies were measured and found to be lower in serum of ovarian cancer patients compared to controls [112].

### 2.9. Cervical Cancer

Thus far, auto-antibodies recognizing two different antigens have been identified in cervical cancer—one is the tetra saccharide CA19-9, which strongly increases in patients with more advanced cervical cancer [113], and one is GAPDH [114]. Interestingly, in this latter study, Xu and colleagues [114] found a negative correlation between the level of circulating auto-antibodies and the severity of the cervical lesions. The authors showed that lower anti-GAPDH IgG levels could discriminate between normal cells and more progressive stages of cervical lesions that often lead to cervical cancer (e.g., normal vs. cervical intraepithelial neoplasia (CIN) II and III). This negative correlation of the GAPDH antibody level with different stages of cervical lesion can function as an important biomarker in cervical cancer, especially when combined with additional markers, such as the presence of antibodies to the human papillomavirus (HPV), a pathogen found in nearly all cervical cancer patients [115].

A combination of ELISAs for anti-CA15-3, anti-CEA, and anti-CA19-9 reliably discriminated CINs from normal cases, and cancer from normal cases, suggesting that this combination assay could be useful for primary screening of cervical cancer [113].

### 2.10. Bladder Cancer

While bladder cancer (BC) is often not perceived as such, it is ranked sixth in absolute incidence rate for men worldwide, while only seventeenth for women [116]. The USA and Europe see the highest incidence rates often related to exposure risk factors such as cigarette smoking, causative of up to 50% of all BC diagnosed as well as some environmental chemical carcinogens [117]. Most BC patients are initially diagnosed with a better-treatable form defined as non-muscle-invasive disease (NMIBC). Several cancer progression subclassifications (using three stages and two grades) have been introduced to define the chance of recurrence, which, in more progressive cases, is very high (50–70%) and will eventually lead to a muscle-invasive disease (MIBC) variant that is associated with a very poor prognosis [117]. For this reason, adequate biomarkers are also needed in bladder cancer in order to identify the aggressive high-grade MIBC. Minami and colleagues [118] found that the expression levels of an important serine/threonine-specific protein phosphatase, PPP1CA, involved in signal transduction, apoptosis, protein synthesis, and cell-cycle regulation, was associated with poorer prognosis, and the authors subsequently investigated IgG serum levels raised against this protein. Serum levels of anti-PPP1CA IgG were found to be higher in BC patients than in healthy individuals, with a specificity of 64.2% and a sensitivity of 65.7%, and were associated with muscular invasion, a higher tumor grade, and poorer prognosis, making it a potential valuable diagnostic and prognostic marker. Besides a role as diagnostic biomarker, understanding the role PP1Ca plays specifically in cancer progression could lead to new therapeutic interventions for this type of cancer [119].

### 2.11. Prostate Cancer

In a study by Tan et al. [120], the proteins SPARC and Fetuin-A were selected for analysis of auto-antibodies as they were shown to be highly expressed at late stages of prostate cancer. Sera from 117 Caucasian American (CA) and 111 African American (AA) prostate cancer patients with Gleason grades 6–10, and healthy controls (CA, *n* = 52; AA, *n* = 45) were analyzed in addition to sera from a biopsy cohort (*n* = 99). The specificity of auto-antibodies against the respective target proteins was confirmed by immunoblot analysis. Both SPARC and Fetuin-A antibodies were detected in the sera, with significantly lower levels in both CA and AA prostate cancer patients compared to healthy controls. The range of auto-antibodies reactivity to SPARC and Fetuin-A was similar in both CA and AA prostate cancer patients, indicating a similar behavior also across ethnic groups [120].

### 2.12. Testicular Seminoma

Testicular seminoma accounts for 40% of testicular cancers and originates from the germinative components of testicular epithelium. Testicular seminoma is highly treatable and has a good prognosis. Diagnosis is mainly made by ultrasound investigation. Often, the onset of a testicular disease is accompanied by paraneoplastic encephalitis and, indeed, almost 20% of patients with paraneoplastic limbic encephalitis have concomitant testicular seminoma [121]. Several studies have shown the presence of serum auto-antibodies in patients with paraneoplastic encephalitis, of which, presently, six are recognized as indicative of an ongoing paraneoplastic encephalitis [122]. In particular, high levels of anti-Ta antibodies in men are associated with testicular seminoma diagnosis. Anti-Ta antibodies react with the paraneoplastic protein PNMA2 (former antibody name: anti-Ma2) [123]. In many cases, anti-Ta antibodies can be detected several months before diagnosis [124]. Mandel-Brem [125] and, more recently, Maudes et al. [126] have described the association of detection of Kelch-like protein 11 auto-antibodies to testicular seminoma in patients with paraneoplastic encephalitis. The presence of elevated concentrations of serum anti-Kelch-like protein 11 auto-antibodies, alone or in combination with anti-Ma auto-antibodies, was associated with a poor response to treatment of testicular seminoma [126].

### 2.13. Lymphomas

Non-Hodgkin’s lymphoma (NHL) is the most common form of lymphoma, a cancer that affects the lymphatic system. A 2012 study investigating the role of the immune system in the progression of NHL found significant levels of circulating auto-antibodies in 150 patients, 50% of which had large B cell lymphoma. The patients were either newly diagnosed, received treatment (i.e., chemotherapy), or were disease-free during follow up. In total, 84% of the patients had one or more auto-antibodies. The antigens identified were Jo-1 and Jo-3, single strand DNA (ssDNA), perinuclear anti-neutrophil cytoplasmic (p-ANCA), antinuclear (ANA), and rheumatoid factor (RF). Patients with newly diagnosed NHL had significantly higher levels of anti Scl-70, anti Jo-1, and RF compared to other patients, indicating the auto-antibodies could not only be used for diagnosis, but could potentially also help with staging [127]. Interestingly, the detected levels of anti-double strand DNA (dsDNA) and anti-ssDNA were relatively low compared to another study by Swissa et al., who showed high levels in 23.8% NHL patients as well as anti-RNP and anti-SM antibodies that were not detected in most of the control group [128]. An extensive investigation into auto-antibodies and recognized neoantigens in human mantle cell lymphomas (MCLs), a rare type of non-Hodgkin lymphoma, was undertaken by Khodadoust et al. [129] using a very thorough genomic and proteomic approach combining MHC isolation, peptide identification, and exome sequencing [129]. However, since no control group of healthy patients was included, no information could be given on diagnostic sensitivity or specificity using these auto-antibody levels. Interestingly, all identified neoantigens were peptides exclusively derived from the lymphoma immunoglobulin heavy or light chain variable regions and, surprisingly, no neoantigens were identified from non-immunoglobulin somatically mutated genes. The study showed that many of the genes presented by MHC were shared between the 17 patients used for the study. However, the peptides were generally unique to each patient except for those with similar MHC-I and/or MHC-II alleles. The authors concluded that, although they could not completely exclude a certain experimental bias towards more abundant epitopes that might have not allowed identification of very rare epitopes, the possibility to identify lymphoma-specific CD4 neoantigens and their use to select and expand endogenous T cells could present an effective lymphoma immunotherapy.

### 2.14. Melanoma

Different from lymphomas and many other cancers, melanomas are often characterized by a very high mutational load, which increases the likelihood of this type of tumor generating neoantigens. In melanomas, the prognostic value of the mutational load has been investigated and was found to be associated with the clinical benefits of immunotherapy such as the adoptive T cell therapy (ACT), which has shown remarkable results in clinical trials in certain patient groups while failing in others [130]. A comprehensive genomic analysis of melanoma tumor samples by Lauss et al. [131] showed a strong positive correlation between a high mutational load and improved clinical outcome following ACT. This also aligns well with the observation that tumors with a high mutational load respond better to treatment with immune checkpoint inhibitors in melanoma and lung cancer [132,133], and is indicative of an increased production of neoantigens. From these and other studies it has also become clear that the presentation of tumor antigens is dependent on the MHC class I antigen processing pathway and subsequent recognition by CD8^+^ T cells. Loss of MHC I antigen presentation was found in many advanced-stage melanomas while high-level expression of MHC class I antigen processing machinery (APM) genes was associated with clinical benefits to immune therapies. When combined, the studies on predictive outcome from ACT seems to indicate that the neoantigen load, predicted by the measured mutational load, can function as an independent predictor for survival after ACT. Two more important melanoma characteristics that were found to be strongly associated with improved outcome from immune therapy were a low-proliferative status and the upregulation of genes responsible for antigen presentation by the cells [132,133].

Due to the high mutational load and consequential formation of neoantigens, circulating auto-antibodies could function as important biomarkers for early detection, incredibly crucial for treatment of this disease. Using protein microarrays, melanoma patient sera was compared with healthy control sera. Antibodies in the patient sera (124 patients) recognized a total of 748 antigens, of which 139 stood out, giving a mean specificity of 97% [134]. Of the 139 antigens, 20 were not reactive with healthy control serum. Interestingly, most of the seroreactive proteins were found to be intracellular (101/139), many of which were nuclear (88/139). Many of the top 139 identified reactive antigens belong to common cancer pathways involved in apoptosis, cell cycle, p53 signaling, MAPK signaling, and immune response. The authors concluded by stating that the detection of combinations of auto-antibodies gave superior sensitivity and specificity, and for their cohort, a set of 10 auto-antibodies gave 79% sensitivity and 84% specificity, much higher than can be accomplished with existing individual biomarkers. A further combination of 10 autoantibody biomarkers (ZBTB7B, PRKCH, p53TP53, PCTK1, PQBP1, UBE2V1, IRF4, MAPK8_tv2, MSN, and TPM1) displayed a sensitivity of 79% and specificity of 84% for primary melanoma detection [134].

With high-throughput screening technologies, it is easy to envision that screening for larger sets of auto-antibodies will further increase the accuracy of diagnostic testing.

### 2.15. Angiosarcoma

Angiosarcoma is a rare soft tissue sarcoma, accounting for less than 2% of total soft tissue sarcomas. Non-specific symptomatology and manifestation, together with the high aggressiveness and propensity of spreading of this type of cancer, make angiosarcomas challenging both for diagnosis and for treatment [135]. Angiosarcoma is often positive for over-expression of VEGF and its receptor (VEGFR), both of which can be the possible target for therapy based on anti-angiogenic agents [135]. Diagnosis of the symptoms and location of angiosarcoma is quite challenging, but recently an association between angiosarcomas and anti-p53 serum auto-antibody levels has been proved statistically relevant and can be exploited both for early diagnosis and for disease staging [136].

### 2.16. Antibody-Mediated Paraneoplastic Syndromes

There are several paraneoplastic syndromes associated to auto-antibodies. Many of them are due to onco-neural auto-antibodies [137], whose molecular and pathogenic relationship with the original cancer is often difficult to identify. Anti-Ri antibodies [138], breast cancer anti-phospholipids antibodies [139,140], cerebellar degeneration [141,142], glomerular diseases [143], and lung cancer [144] are all conditions which still require a proper pathogenic explanation.

To summarize, Table 1 presents the antigens mentioned in the above paragraphs in alphabetical order and Table 2 lists the main auto-antibody combinations that seem promising in early cancer detection or in the prediction of cancer prognosis. Where available, sensitivity and specificity were also included in dedicated columns.

## 3. Discussion

From the data reported above, it is clear that the panel of auto-antibodies that can be detected in cancer patients is often large and still expanding and, with it, a number of observations can be made and questions raised.

The first point to notice is that, in general, the overall number of tumor antigens able to induce an antibody response is surprisingly high (120 reported only in this review, Table 1), indicating the capability of the immune system to detect many native proteins as non-self. It would be interesting to understand the proportion between TAAs and TSAs among them, but the sequence analysis of specific genes encoding for neoantigens in solid cancer is still challenging and has not been systematically incorporated in many of the studies reported. At the same time, comprehensive genomic profiling has not been associated with antibody detection in such studies either. Thus, while mutations of some common antigens, such as p53, have been well documented, much less is known about the mutational status of the newly identified auto-antigens, leaving open the question of their true nature. The results of necessary further studies might even challenge the very same concepts and definitions of TSA and TAA and make the auto-antigenic situation in cancer quite similar to the one detected in rheumatologic diseases, where a general pathogenic mechanism for the development of auto-antibodies has not yet been identified as well. A feature shared by the two conditions is represented by inflammation, whose potential causative correlation with cancer was identified already in the 19th century [148]. The inflammatory cells and molecules present in the tumor microenvironment might form the basis to explain the appearance of the antibody response towards selected antigens. In this respect, increasing knowledge about the mechanisms and differences between TSAs and TAAs in cancer will likely help to understand pathogenic mechanisms in autoimmune diseases as well.

Related to this, two important concepts need to be understood. One regards the mechanisms involved in MHC-mediated presentation of cancer auto-antigens and the trigger that primes an immune response towards them, both very challenging to investigate. The fact that a similar MHC-I was identified in patients showing similar epitope peptides in auto-antibodies detected in lymphomas [129] suggests that this step is, indeed, a relevant factor for auto-antibodies induction. However, since lymphomas are themselves tumors of the immune system, they might represent the exception more than the rule.

A second relevant point is that the prevalence of the antibody response in most of the cancer patient groups analyzed is relatively low, and, in fact, efficient detection of an early cancer is more successful when panels of auto-antibodies or combinations of auto-antibodies and other biomarkers are analyzed, instead of single antibodies, as testified by the success of the EarlyCDT-Lung assay. The reasons why only subgroups of patients mount an immune response and only towards certain antigens is still unknown and will require genome-wide sequencing studies, for example those focused on Human Leucocyte Antigen (HLA) associations, to understand if there is an underlying genetic basis.

A third point to consider is that what is often defined as a “single” antibody detected by ELISA or similar tests against an antigen, is in fact likely to be a signal generated by multiple antibodies derived from a polyclonal population of B cells. This is the scenario that is emerging in the field of auto-immune diseases [149], and dissecting this complexity will not be easy. The current availability of suitable techniques allowing isolation of single antibody specificities towards specific antigens from human B cells will be central in this task.

A fourth and key point is that it is now clear that the action of the auto-antibodies against cancer antigens can be either protective against or supportive towards the cancer. For example, the now well-understood mechanism underlying the role of anti-HSPA4 IgGs in breast cancer metastasis [100] represents a turning point paradigm. Auto-antibodies might initially represent a simple epiphenomenon or a by-product of the inflammatory microenvironment but could become a main player in the tumor progression, conditioning its evolution and outcome.

As a fifth point, it has been reported that the appearance of certain auto-antibody species occurs only after chemotherapy and, because of this, it can represent a useful prognostic parameter [89]. Identifying the underlying mechanisms, such as the release of cytoplasmic and nuclear proteins after induced cell death for many of such antibody species is an important task to be undertaken and will also require a deeper understanding of the cancer microenvironment.

As a sixth and final point, it is not irrelevant to notice that the presence of anti-cancer auto-antibodies can influence the detection of their cognate antigen, as shown in the case of CA125 [102], indicating an immediate practical implication in tumor marker detection.

A frequent limitation encountered in the studies here collected was the lack of sensitivity and specificity data (See Table 1 and Table 2, “N.A.”). Very often this was due to the fact that they were not specifically designed for this purpose, being early investigational approaches. In some studies, independent validation groups were also not available. Those for which they were reported are highlighted in bold in Table 2.

## 4. Conclusions

The identification of many disease-specific serum biomarkers that can be measured with relatively non-invasive methods has provided accurate diagnostic tools for many different diseases, most importantly cancer, for which early detection is essential for effective treatment. While auto-antibodies have been well-studied in autoimmune diseases, they have only recently been recognized as being potential tools for diagnostic, staging, or prognostic application in cancer. Auto-antibodies are found against normal proteins overexpressed on, or released from, tumor cells or against mutated proteins, (i.e. neoantigens), but only in some cases their function has been determined at the molecular level. To fully benefit from the use of these auto-antibodies, it is essential to increase our understanding on the source of the newly recognized antigens and the process by which the immune response is provoked. There are therefore several issues that need to be tackled with further research in the field of anti-cancer auto-antibodies. Out of these, three are related to the complex molecular nature of the auto-antibody response against cancer antigens. Within these polyclonal mixtures, in fact, groups of single specificities are likely to target different epitopes, with different, in some cases possibly even opposite, effects on cancer development. In this perspective, a high priority should be given to (i) dissecting the complex molecular nature of the auto-antibody response against cancer antigens to identify the most immunogenic epitopes, (ii) defining the details of the pathogenic role of each auto-antibody specificity in cancer promotion or suppression, and (iii) further extending the use of anti-cancer auto-antibodies in early detection and prognosis of cancer. Understanding these key points will help in exploiting the natural response of our body against cancer auto-antigens in a patient-specific cancer-tailored treatment, presenting a true personalized medicine approach.

## Figures and Tables

**Table 1 cancers-13-00813-t001:** Non-comprehensive list of antigens involved in auto-antibody production in cancer.

Antigen (Short Name/Gene Name)	Antigen Name	Uniprot ID	Cancer	I/E *,^§^	Sensitivity	Specificity	Notes
14-3-3ζ	14-3-3 protein zeta	P29310	Gastric cancer	I	22.58%	92.26%	
ACTR3	Actin-related protein 3	P61158	Lung cancer	I	20.7%	>90%	Early stage marker
AEG-1	Protein Lyric	Q86UE4	Gastrointestinal cancer	I	59.1%	100%	Stage-related (late-stage patients)
AHSG	Alpha-2-HS-glycoprotein (Fetuin-A)	P02765	Prostate cancer	E	N.A.	N.A.	
ALDH1B1	Aldehyde dehydrogenase X	P30837	Colorectal cancer	I	62.31–75.68%	73.78–63.06%	
ALK	Anaplastic lymphoma kinase	Q9UM73	Lung cancer	E	N.A.	N.A.	NSCLC inversely correlated with stage of disease
ALMS1	Alstrom syndrome protein 1	Q8TCU4	Lung cancer	I	N.A.	N.A.	
Annexin-1	Annexin A1	P04083	Lung cancer	E	N.A.	N.A.	
			Colorectal cancer		2.5%	N.A.	
ATP1A4	Sodium/potassium-transporting ATPase subunit alpha-4	Q13733	Lung cancer	E	N.A.	N.A.	
BCL7A	B cell CLL/lymphoma 7 protein family member A	Q4VC05	Lung cancer	N.A.	30.9%	94.3%	IgA autoantigen, early stage marker
BCOR	BCL-6 corepressor	Q6W2J9	Ovarian cancer	I	73%	>94%	
C1D	Nuclear nucleic acid-binding protein C1D	Q13901	Ovarian cancer	I	N.A.	N.A.	
C12orf54	Uncharacterized protein	Q6X4T0	Lung cancer	N.A.	N.A.	N.A.	
CA125	Mucin-16 (Cancer Antigen 125)	Q8WXI7	Lung cancer	E	N.A.	N.A.	
			Ovarian cancer		95%	40%	
CA19-9	Tetrasaccharide CA19-9	N.A.	Cervical cancer	N.A.	3.2%	N.A.	
CAGE	Cancer-associated gene 1 protein	Q8TC20	Lung cancer	N.A.	N.A.	N.A.	
CAMSAP2	Calmodulin-regulated spectrin-associated protein 2	Q08AD1	Lung cancer	I	N.A.	N.A.	
CCDC155	Coiled-coil domain-containing protein 155	Q8N6L0	Ovarian cancer	I	95%	40%	Early ovarian cancer
CCL18	C-C motif chemokine 18	P55774	Ovarian cancer	E	N.A.	N.A.	
CCNB1	G2/mitotic-specific cyclin-B1	P14635	Central nervous system (CNS) tumors	I	10.6%	96.2%	
CD25	Interleukin-2 receptor subunit alpha	P01589	Lung cancer	E	N.A.	N.A.	
CEA	Carcinoembryonic antigen-related cell adhesion molecule 5	P06731	Lung cancer	E	N.A.	N.A.	(Antigen only)
CENPF	Centromere protein F	P49454	Colorectal cancer	I	64.34%	67.27%	Colorectal cancer
					62.67%	62.67%	Advanced adenoma
CREB3	Cyclic AMP-responsive element-binding protein 3	O43889	Ovarian cancer	I	87%	98%	
CRYM	Ketimine reductase mu-crystallin	Q14894	Central nervous system (CNS) tumors	I	N.A.	N.A.	Downregulated in menangiomas
CT47A	Cancer/testis antigen 47A	Q5JQC4	Lung cancer	N.A.	N.A.	N.A.	
CTAG1	Cancer/testis antigen 1	P78358	Gastric cancer	I	N.A.	N.A.	
			Colorectal cancer		64.62–59.46%	70.27–56.36%	Colorectal cancer, advanced adenoma
			Lung cancer		23.5%	97.7%	IgG autoantigen, early stage marker
CTAG2	Cancer/testis antigen 2	O75638	Gastrointestinal cancer	I	16.6%	99.5%	Not stage-related
CXCL1	Growth-regulated alpha protein	P09341	Ovarian cancer	E	N.A.	N.A.	
Cyclin B1	G2/mitotic-specific cyclin-B1	P14635	Colorectal cancer	I	15.6–32.7%	97.6%	Hepatocarcinoma
Cytokeratin 20	Keratin, type I cytoskeletal 20	P35900	Lung cancer	I	N.A.	N.A.	
DDX4	Probable ATP-dependent RNA helicase DDX4	Q9NQI0	Lung cancer	I	25.0%	96.6%	IgG autoantigenearly-stage marker
DDX53	Probable ATP-dependent RNA helicase DDX53	Q86TM3	Gastrointestinal cancer	I	6.8%	100%	Not stage-related
EFCAB2	Dynein regulatory complex protein 8	Q5VUJ9	Central nervous system (CNS) tumors	I	N.A.	N.A.	
ENO1	Alpha-enolase	P06733	Gastric cancer	I	N.A.	N.A.	
ERP44	Endoplasmic reticulum resident protein 44	Q9BS26	Colorectal cancer	I	40%	100%	
FSCN1	Fascin	Q16658	Esophageal cancer	I	N.A.	N.A.	
FATE1	Fetal and adult testis-expressed transcript protein	Q969F0	Adrenocortical carcinoma	I	N.A.	N.A.	
FXR1	Fragile X mental retardation syndrome-related protein 1	P51114	Ovarian cancer	I	N.A.	N.A.	
GAGE7	G antigen 7	O76087	Lung cancer	N.A.	N.A.	N.A.	
Galectin1	Galectin1	P09382	Colorectal cancer	E	11%	N.A.	
GAPDH	Glyceraldehyde-3-phosphate dehydrogenase	P16858	Cervical cancer	I	80%	96.6%	
GBU4-5	Putative ATP-dependent RNA helicase TDRD12	Q587J7	Lung cancer	I	N.A.	N.A.	Also known as FLI3072
GCC2	GRIP and coiled-coil domain-containing protein 2	Q8IWJ2	Lung cancer	I	N.A.	N.A.	
GNA11	Guanine nucleotide-binding protein subunit alpha-11	P29992	Liver cancer	I	N.A.	N.A.	Hepatocarcinoma
GNAS	Guanine nucleotide-binding protein G(s) subunit alpha isoforms short	P63092	Liver cancer	E	48.4%	>90%	Hepatocarcinoma
GPR78	G-protein coupled receptor 78	Q96P69	Gastrointestinal cancer	E	28.3%	100%	Non stage-related
GREM1	Gremlin-1	O60565	Lung cancer	E	N.A.	N.A.	
GRP78	Glucose regulated protein 78	P11021	Ovarian cancer	I	N.A.	N.A.	
HCA25a	Hepatocellular carcinoma-associated antigen HCA25a	Q8NHH4	Colorectal cancer	N.A.	3%	N.A.	
HCC-22-5 (SMP30)	Senescence marker protein-30 (Regucalcin)	Q15493	Colorectal cancer	I	4%	N.A.	
HCC1	Protein SCO1 homolog 1	Q8VYP0	Liver cancer	I	N.A.	N.A.	Hepatocarcinoma
HDAC7A	Histone deacetylase 7	Q8WUI4	CNS tumors	I	N.A.	N.A.	
HE4	Human epididymis secretory protein 4	Q14508	Lung cancer	E	67.21%	96.23%	
HMGB3	High mobility group protein B3	O15347	Lung cancer	I	N.A.	N.A.	
HOXA7	Homeobox protein Hox-A7	P31268	Ovarian cancer	I	66.7%	100%	
HRNR	Hornerin	Q86YZ3	Lung cancer	I	N.A.	N.A.	
HSF1	Heat shock factor protein 1	Q00613	Ovarian cancer	I	95%	80%	Early ovarian cancer
Hspa4	Heat shock 70 kDa protein 4	P34932	Breast cancer	I	N.A.	N.A.	
Hsp40	DnaJ homolog subfamily B member 1	P25685	Colorectal cancer	I	7%	N.A.	
Hsp60	Heat shock protein 60	P10809	Breast cancer	I	N.A.	N.A.	31% cases of early breast cancer, 32.6% ductal carcinoma in situ
			Lung cancer		N.A.	N.A.	
Hsp70	Heat shock 70 kDa protein	P0DMV8	Colorectal cancer	I	12%	N.A.	
			Esophageal cancer		N.A.	N.A.	
HuD *	ELAV-like protein 4	P26378	Lung cancer	I	N.A.	N.A.	
IGF2BP2	Insulin-like growth factor 2 mRNA-binding protein 2	Q9Y6M1	Gastric cancer	I	N.A.	N.A.	
IGHG4	Immunoglobulin heavy constant gamma 4	P01861	Central nervous system (CNS) tumors	E	N.A.	N.A.	
IL8	Interleukin-8	P10145	Ovarian cancer	E	65.5%	98%	Stage I–II
IMP1	Insulin-like growth factor 2 mRNA-binding protein 1	Q9NZI8	Colorectal cancer	I	13.3–21.7%	97.6–100%	
Kelch-like protein 11 *	Kelch-like protein 11	Q9NVR0	Testicular seminoma	I	N.A.	N.A.	Patients with paraneoplastic encephalitis; associated with poor response to treatment
KIF13B	Kinesin-like protein	Q9NQT8	Lung cancer	I	N.A.	N.A.	
KM-HN-1	Coiled-coil domain-containing protein 110	Q8TBZ0	Colorectal cancer	I	9%	N.A.	
Koc	Insulin-like growth factor 2 mRNA-binding protein 3	O00425	Colorectal cancer	I	8.9–15.2%	98.8–100%	
LIN28B	Protein lin-28 homolog B	Q6ZN17	Lung cancer	I	N.A.	N.A.	
Lmyc2	Protein L-Myc	P12524	Lung cancer	I	N.A.	N.A.	
Ma *	Paraneoplastic antigen Ma1	Q8ND90	Testicular seminoma	I	N.A.	N.A.	
	Paraneoplastic antigen Ma2	Q9UL42	Testicular seminoma	I	N.A.	N.A.	
MAGEA1	Melanoma-associated antigen 1	P43355	Lung cancer	I	N.A.	N.A.	
MAGEA4	Melanoma-associated antigen 4	P43358	Lung cancer	N.A.	N.A.	N.A.	
MAGEA3	Melanoma-associated antigen 3	P43357	Gastrointestinal cancer	I	3.4%	100%	Not stage-related
MAGEB1	Melanoma-associated antigen B1	P43366	Lung canceer	N.A.	N.A.	N.A.	
MAGEC1	Melanoma-associated antigen C1	O60732	Gastrointestinal cancer	N.A.	3.4%	100%	Not stage-related
MAGEC2	Melanoma-associated antigen C2	Q9UBF1	Lung cancer	I	27.9%	95.4%	IgG autoantigen, early stage marker
MDM2	E3 ubiquitin-protein ligase	Q00987	Gastric cancer	I	N.A.	N.A.	
MED14	Mediator of RNA polymerase II transcription subunit	O60244	Lung cancer	I	N.A.	N.A.	
MGMT	Methylated DNA protein cysteine methyltransferase	P16455	Central nervous system (CNS) tumors	I	N.A.	N.A.	Associated with higher risk of chemotherapy resistance and disease recurrence; positive rate: MGMT-02 45%, MGMT-04 27%, MGMT-07 21%, MGMT-10 13%, MGMT-18 24%
MMP-7	Matrilysin	P09237	Esophageal cancer	E	N.A.	N.A.	
MNRR1 (CHCHD2)	Mitochondrial-nuclear retrograde regulator 1	Q9Y6H1	Breast cancer	I	N.A.	N.A.	Metastasis and aggressive tumors
MPB-1	Alpha-enolase	P06733	Lung cancer	I	N.A.	N.A.	
MRPL46	39S ribosomal protein L46, mitochondrial	Q9H2W6	Ovarian cancer	I	73%	>94%	
MSH2	DNA mismatch repair protein Msh2	P43246	Liver cancer	I	42.1%	>90%	Hepatocarcinoma
MTERF4	Transcription termination factor 4	Q7Z6M4	Lung cancer	I	33.5%	96.6%	IgA autoantigen, early stage marker
MUC1	Mucin 1	P15941	Lung cancer	E	N.A.	N.A	Mucin 1 subunit ß is translocated in the nucleus
MUC17	Mucn-17	Q685J3	Lung cancer	I/E	N.A.	N.A.	
Myc	Myc proto-oncogene protein	P01106	Gastric cancer	I	33.17%	90.17%	
			Colorectal cancer		4.4–21.7%	94.8–100%	
			Liver cancer		9%	N.A.	
			Ovarian cancer		N.A.	N.A.	
NPM1	Nucleophosmin	P06748	Gastric cancer	I	N.A.	N.A.	
NUDT11	Diphosphoinositol polyphosphate phosphohydrolase 3-beta	Q96G61	Ovarian cancer	I	32.4%	100%	Serous ovarian cancer
NY-ESO-1 *	Cancer/testis antigen 1	P78358	Liver cancer	I	10%	N.A.	Hepatocarcinoma Intrahepatic cholangio-carcinoma
			Lung cancer		N.A.	N.A.	
			Colorectal		8%	N.A.	
			Esophageal cancer		31%	N.A.	
p16	Cyclin-dependent kinase inhibitor 2A	P42771	Esophageal cancer	I	N.A.	N.A.	
			Gastric cancer		N.A.	N.A.	
			Liver cancer		N.A.	N.A.	
p53	Cellular tumor antigen p53	P04637	Breast cancer	I	N.A.	N.A.	Triple negative
			Gastric cancer		N.A.	N.A.	In combination with CEA and/or CA19-9, associated with lymphatic node and distant metastasis
			Colorectal cancer		16%	N.A.	
			Liver cancer		11%	N.A.	Hepatocarcinoma
			Lung cancer		N.A.	N.A.	EarlyCTD Lung panel, non-small cell lung cancer (NSCLC)
			Angiosarcoma		N.A.	N.A.	
			Ovarian cancer		N.A.	N.A.	
p62	Sequestosome-1	Q13501	Gastric cancer	I	N.A.	N.A.	
			Colorectal cancer		11.1–23.4%	97.1–98.8%	
			Liver cancer		18%	N.A.	Hepatocarcinoma
p90	N.A.	G8IFA7	Colorectal cancer	N.A.	7%	N.A.	
			Liver cancer	N.A.	N.A.	N.A.	Hepatocarcinoma
PAGE3	P antigen family member 3	Q5JUK9	Lung cancer	N.A.	N.A.	N.A.	
PAX5	Paired box protein Pax-5	Q02548	Liver cancer	I	N.A.	N.A.	Hepatocarcinoma
PGP9.5	Ubiquitin carboxyl-terminal hydrolase isozyme L1	P09936	Lung cancer	I	N.A.	N.A.	
PI3K	Phosphatidylinositol 4,5-bisphosphate 3-kinase	P42336	Breast cancer	I	N.A.	N.A.	Triple negative
PPP1C	Serine/threonine-protein phosphatase PP1-alpha catalytic subunit	P62136	Bladder cancer	I	64.2%	65.7%	
PrxVI	Peroxiredoxin-6	P30041	Colorectal cancer	I	4%	N.A.	
PSIP1	PC4 and SFRS1-interacting protein	O75475	Lung cancer	I	32.07%	>90%	Early stage marker
PTCH1	Protein patched homolog 1	Q13635	Liver cancer	E	N.A.	N.A.	Hepatocarcinoma
PTEN	Phosphatidyl-inositol 3,4,5-trisphosphate 3-phosphatase and dual-specificity protein phosphatase	P60484	Gastric cancer	I/E	N.A.	N.A.	
PVR	Poliovirus receptor	P15151	Ovarian cancer	E	17.6%	96.8%	Serous ovarian cancer
RalA	Ras-related protein Ral-A	P11233	Gastric cancer	E	15%	N.A.	In combination with CEA and/or CA19-9
			Colorectal cancer		14%	N.A.	
			Liver cancer		17%	N.A.	Hepatocarcinoma
RhoGDI	Rho GDP-dissociation inhibitor 1	P52565	Ovarian cancer	I	89.5%	80%	Ovarian cancer
RPS6KA5	Ribosomal protein S6 kinase alpha-5	O75582	Lung cancer	I	15.4%	>90%	Early stage marker
SMG1	Serine/threonine-protein kinase	Q96Q15	Lung cancer	I	N.A.	N.A.	
SOX2	Transcription factor SOX-2	P48431	Lung cancer	I	N.A.	N.A.	EarlyCTD Lung panel, non-small cell lung cancer (NSCLC)
SP17	Sperm surface protein Sp17	Q15506	Head and neck cancers (HCN)	E	N.A.	N.A.	
SPAG9	Sperm-associated antigen	O60271	Liver cancer	I	71.0%	87.3%	Hepatocarcinoma
SPARC	SPARC	P09486	Prostate cancer	E	N.A.	N.A.	
STAT6	Signal transducer and activator of transcription 6	P42226	CNS tumors	I	N.A.	N.A.	
Sui1	Eukaryotic translation initiation factor 1	P41567	Liver cancer	I	19%	N.A.	Hepatocarcinoma
Survivin	Baculoviral IAP repeat-containing protein 5	O15392	Colorectal cancer	I	4.4–56.9%	64.1–100%	
			Liver cancer		42.4%	>90%	Hepatocarcinoma
Ta proteins *	Tail-anchored proteins	N/A	Testicular seminoma	E	N.A.	N.A.	[145]
TALDO1	Transaldolase	P37837	Colorectal cancer	I	56.9%	66.7%	
Tg	Thyroglobulin	P01266	Differentiated thyroid carcinoma (DTC)	E	N.A.	N.A.	
TIMELESS	Protein timeless homolog	Q9UNS1	Lung cancer	I	N.A.	N.A.	
TIZ (ZNF675)	Zinc finger protein 675	Q8TD23	Ovarian cancer	I	N.A.	N.A.	
TM4SF1	Transmembrane 4 L6 family member 1	P30408	Ovarian cancer	E	N.A.	N.A.	
TNS1	Tensin-1	Q9HBL0	Lung cancer	I	N.A.	N.A.	
TOP1	DNA topoisomerase 1 fragment	P11387	Esophageal cancer	I	61.8%	100%	TOP1 fragment (329–765)
TOP2A	DNA topoisomerase 2-alpha	P11388	Lung cancer	I	24.5%	>90%	Early stage marker
TRIM21 *	E3 ubiquitin-protein ligase TRIM21	P19474	Ovarian cancer	I	33%	100%	
TRIM33 *	E3 ubiquitin-protein ligase TRIM33	Q9UPN9	Lung cancer	I	32.4%	94.3%	IgA autoantigen early stage marker
TRIM39	E3 ubiquitin-protein ligase TRIM39	Q9HCM9	Ovarian cancer	I	14.7%	96.8%	Serous ovarian cancer
TUBA1C	Tubulin alpha-1C chain	Q9BQE3	Ovarian cancer	I	89%	75%	
UQCRC1	Cytochrome b-c1 complex subunit 1	P31930	Colorectal cancer	I	57.7%	70.27%	Colorectal cancer
					86.49%	27.93%	Advanced adenoma
VCX	Variable charge X-linked protein 1	Q9H320	Lung cancer	I	N.A.	N.A.	
VEGF	Vascular endothelial growth factor	P15692	Colorectal concer	E	4%	N.A.	
VEGFR1	Vascular endothelial growth factor receptor 1	P17948	Lung cancer	E	N.A.	N.A.	

* indicates paraneoplastic antigens. ^§^ E/I: Intracellular versus Extracellular localization.

**Table 2 cancers-13-00813-t002:** Autoantibody panels useful in cancer diagnosis and treatment.

Panel (Antigens)	Cancer	Sensitivity	Specificity	Reference
c-Myc, p16, HSPD1, PTEN, p53, NPM1, ENO1, p62, HCC1.4	Gastric cancer	71.5%	71.3%	[36]
p53, heat shock protein 70, HCC-22-5, peroxiredoxin VI, KM-HN-1, p90 TAA	Gastric cancer	49%	92.4%	[38]
14-3-3 zeta, CEA, CA199, CA724	Gastric cancer	52.7%	N.A.	[41]
p62, c-Myc, NPM1, 14-3-3ξ, MDM2, p16	Gastric cancer	78.92%	74.7	[36]
CTAG1B/CTAG2, DDX53, IGF2BP2, p53, MAGEA3	Gastric cancer	21%	91%	[37]
RalA, CEA, CA19-9	Gastric cancer	N.A.	N.A.	[33]
p53, CEA, CA19-9	Gastric cancer	N.A.	N.A.	[33]
**c-MYC, cyclin B1, p62, Koc, IMP1, survivin**	**Colorectal cancer**	**15.5–88%**	**71.4–100%**	**[42]**
**ALDH1B1, UQCRC1, CTAG1, CENPF**	**Colorectal cancer**	**75.68%**	**73.87%**	**[39]**
p53, RalA, HSP70, Galectin1, KM-HN-1	Colorectal cancer	56%	85%	[46]
p53, RalA, HSP70, Galectin1, KM-HN-1, NY-ESO-1, p90, Sui1, HSP40, Cyclin B1, HCC-22-5, c-myc, PrxVI, VEGF, HCA25a, p62, Annexin	Colorectal cancer	N.A.	N.A.	[46]
SNX1, MYLK, VDAC1, IGHG1, CCDC32, EYA1, CD44, NOL3, PQBP1, EXOSC7	Glioma	88%	87%	[146]
C14orf80, GCK, HSD17B14, LYPLAL1, MAGEA4, MLX, RTN4, SNX1, TEX264, ARHGAP17	Glioma	89%	100%	[146]
PGM2, DR1, HIBADH, PGM2, DR1, HIBADH	Glioma	77%	95%	[146]
**Sui1, p62, RalA, p53, NY-ESO-1, c-myc**	**Liver cancer**	**56%**	**91%**	**[50]**
**PTCH1, GNA11, PAX5, GNAS, MSH2, Survivin, p53**	**Liver cancer**	**N.A.**	**N.A.**	**[53]**
**HCC1, P16, P53, P90, Survivin**	**Liver cancer**	**88%**	**84.1%**	**[54]**
p53, NY-ESO-1, CAGE, GBU4–5, Annexin 1, SOX2	Lung cancer(early CTD-lung)	39%	89%	[73]
CD25, MUC1, VEGFR1	Lung cancer	49.6%	95%	[86]
**p53, NY-ESO-1, CAGE, GBU4–5, Annexin 1, SOX2, Lmyc2 and cytokeratin 20 or alpha-enolase, cytokeratin 20**	**Lung cancer** **(implemented early CTD-lung)**	**41.6%**	**92.1%**	**[72]**
**p53, NY-ESO-1, CAGE, GBU4-5, SOX2, HuD, MAGE A4**	**Lung cancer**	**41%**	**91%**	**[73]**
**P53, SOX2, GAGE 7**	**Non-small cell lung cancer**	**39%**	**81.8%**	**[85]**
**SOX2, GAGE 7, CAGE, MAGE A1, P53, GBU4-5, PGP9.5**	**Non-small cell lung cancer**	**90%**	**57.9%**	**[85]**
**p53, PGP9.5, SOX2, GAGE7, GBU4-5, MAGE A1, CAGE**	**Lung adenocarcinoma with ground-glass nodules (GGNs)**	**48.6%**	**92.7%**	**[84]**
**BCL7A, TRIM33, MTERF4, CTAG1A, DDX4, MAGEC2**	**Lung cancer**	**73.5%**	**>85%**	**[81]**
**CEA, CA125, Annexin A1-Ab, Alpha enolase-Ab**	**Lung cancer**	**86.5%**	**82.3%**	**[87]**
**GREM1, HMGB3, PSIP1**	**Lung cancer**	**58.07%**	**76.71%**	**[88]**
MUC17, CAMSAP2, KIF13B, SMG1, MED14, ALMS1, GCC2, TIMELESS, TNS1, ATP1A4, HRNR	Lung cancer	N.A.	N.A.	[83]
Anti-pituitary (APA), anti-hypothalamus (AHA)	Brain cancer	N.A.	N.A.	[29]
**p53, NY-ESO-1, MMP-7, Hsp70, PRDX6, Bmi-1**	**Esophagogastric junction adenocarcinoma**	**61.4%**	**90%**	**[64]**
HCCR, C-myc, MDM2, miR-21, miR-223, miR-375	Esophageal squamous cell carcinoma	69%	90%	[147]
**TRIM21, NY-ESO-1, p53, PAX8**	**Ovarian**	**46%**	**98%**	**[106]**
C1D, TM4SF, TIZ, FXR1 combined with CCL18 and CXCL1 antigens	Ovarian	73.14%	94.95%	[108]
ICAM3, CTAG2, p53, STYXL1, PVR, POMC, NUDT11, TRIM39, UHMK1, KSR1, NXF3	Ovarian	45%	98%	[111]
CA15-3, CEA, CA19-9	Cervical cancer	90.3%	82.1%	[113]
ZBTB7B, PRKCH, p53, PCTK1, PQBP1, UBE2V1, IRF4, MAPK8_tv2, MSN, TPM1	Melanoma	79%	84%	[134]

In bold: panels validated by an independent validation group are reported.

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
