# Peer review of "Anti-Cancer Auto-Antibodies: Roles, Applications and Open Issues"

_cancers, 2021, doi:10.3390/cancers13040813_

Round 1

Reviewer 1 Report

This a well-organized review of the topic of cancer autoantibodies and their utility in diagnostics and potential in therapeutics. However, the usefulness of the information is diminished by a number of missing elements of the presence of autoantibodies in healthy, benign or pre-neoplastic disease subjects.  Healthy donor groups are mentioned occasionally without any reference to specificity except for ovarian cancer in line 484 and melanoma in line 651. Appropriate studies using these important control subjects are important in the ultimate implementation of autoantibodies in clinical oncology.  The authors must include data on the control subjects for this review to be valuable.  Where there was no data presented in the literature cited on healthy or benign disease subjects, the authors should indicate that this was a limitation of that particular autoantibody or panel of autoantibodies.

Rarely were studies cited in which there is an independent validation set of subject samples.

In the tables of autoantigens and panels of autoantigens the authors must include the sensitivity and specificity for each marker or panel.  Similarly, in the text this data is also missing.  Many of the panels mentioned in the text are not included in the table 2.  This data should be included in table 2 with the overall sensitivity and specificity in order for the readers to be able to know if the panel is valuable for clinical testing.  Those panels that are validated using an independent validation set should be highlighted in table 2.

Frequently paraneoplastic autoantibodies were mentioned without citing that they are paraneoplastic autoantibodies.  This fact need to be built into the two tables as well.

Authors frequently use the term “correlated” when no true correlation was shown in the paper cited.  They should correct those to “associated”

For example, anti-Ma auto-antibodies are said to be correlated with poor response to treatment of seminomas but no correlation coefficient is cited.  Line 382

Specific Issues:

Line 130 typo:  p53 uto-antibodies

Line 209:  “proved” should be changed to “proven”

Line 214:  the author cite work on lymphoma in the lung cancer section.

Line 242:  The patients in the high titer group (> 100 U/mL) showed a worse survival than those in the other groups.  The authors need to state to which antibody they are referring.

Line 264:  Positive rates increased to 56, 62, 66, 71, and 73% using panels of 6, 9, 11, 14, and 17 antibodies---- this highlights the general problem in the review of positivity or sensitivity without referring to specificity or accuracy or even positivity in benign or pre-neoplastic diseases general.

Line 270:  “The work on immunotherapy” needs to be cited in the references.

Line 280:  (phosphatase and tensin homolog) should be defined as the PTEN gene which is it best known name.

Line 283:  a healthy donor group is mentioned but no specificity is defined for this study.

Line 287:  healthy donors are mentioned but no specific references to false positives or specificity of individual or panels of markers.

Line 336-7:  p53 autoantibodies are mentioned as prognostic biomarkers but without any reference to the actual numeric frequencies or the accuracy.

Line 356:  anti-MMP7 is mentioned as “connected to staging and invasiveness [79]”.  How was it connected?  Be more specific.

Lines 489 and 494:  A study is cited in which the discovery set of samples and the validation set seem to be the same.  That does not constitute an unbiased validation.

Lines 560-2:  “The Authors showed that lower anti-GAPDH IgG levels could discriminate between normal cells and more progressive stages of cervical lesions that often lead to cervical cancer (e.g., normal vs. cervical intraepithelial neoplasia (CIN) II and III).”   What is the sensitivity, specificity and accuracy of this marker????

Reviewer 2 Report

Major point,

Although the author stated at the beginning to review the "function" of the autoantibodies, the main documents just focused on the positive rate of those antibodies in each cancer type. Please try to state much more "function" or just delete these sentences on the front page.

Minor point,

  1. Before going into each cancer type, please summarize a few universal antibodies, such as p53, NYESO-1.
  2. The order of the cancer types should be in order with anatomical order from head to foot. 
  3. The number of words should be related to the number of publications in this research field.
  4. Please try to refer to the following papers.
  5. Panel of autoantibodies against multiple tumor-associated antigens for detecting gastric cancer. Hoshino I, Nagata M, Takiguchi N, Nabeya Y, Ikeda A, Yokoi S, Kuwajima A, Tagawa M, Matsushita K, Satoshi Y, et al.Cancer Sci. 2017 Mar;108(3):308-315. doi: 10.1111/cas.13158.
  6. New Assay System Elecsys Anti-p53 to Detect Serum Anti-p53 Antibodies in Esophageal Cancer Patients and Colorectal Cancer Patients: Multi-institutional Study. Yajima S, Suzuki T, Oshima Y, Shiratori F, Funahashi K, Kawai S, Nanki T, Muraoka S, Urita Y, Saida Y, et al
  7. Clinical impact of preoperative serum p53 antibody titers in 1487 patients with surgically treated esophageal squamous cell carcinoma: a multi-institutional study. Takashi S, Satoshi Y, Akihiko O, Naoya Y, Yusuke T, Kentaro M, Yu O, Yasuaki N, Koichi Y, Takashi F, et al.Esophagus. 2021 Jan;18(1):65-71. doi: 10.1007/s10388-020-00761-6. Epub 2020 Jul 26.
  8. Changing pattern of tumor markers in recurrent colorectal cancer patients before surgery to recurrence: serum p53 antibodies, CA19-9 and CEA. Ushigome M, Shimada H, Miura Y, Yoshida K, Kaneko T, Koda T, Nagashima Y, Suzuki T, Kagami S, Funahashi K.Int J Clin Oncol. 2020 Apr;25(4):622-632. doi: 10.1007/s10147-019-01597-6. Epub 2019 Dec 9.PMID: 31820210

Reviewer 3 Report

This is a an excellent review in an area of research cancer immunologists will find very interesting and topical. The authors have provided an extensive documentation (and tabulation) of auto-antibodies and their implications (clinical, diagnosis, prognosis, survival, biomarker, target etc) in a broad range of cancers- both solid and haematological. For each tumour type, they provide a concise narrative explaining the significance, important concepts that may dictate their role in tumour progression or suppression and where relevant, details of clinical studies and findings have been presented in support of their arguments. I have seen/read quite a few studies that have focussed on identification of auto-antibodies as biomarkers but this review provides insights on how they could reveal clues to why autoantigens are rendered immunogenic and the reasons to investigate underlying causes of autoantibody production. Readers will find this review very insightful and timely, and I recommend its publication. 

Reviewer 4 Report

This manuscript summarizes many papers on auto-antibodies in cancer comprehensively. It is so wonderful and useful for other researchers.

Some of the antibody biomarkers are results of high expression in tumor cells, and therefore, can be used for early detection, diagnosis and monitoring of cancer. Some of the antibodies may have a pathogenic role in cancer promotion or suppression. I understand that there are variable types and roles of antibody biomarkers. I wonder, what do you want to say in this review? Did you find some general phenomenon or characteristics in the markers? For example, antibodies which recognize intracellular proteins may be the results of high expression in cancer cells whereas those which recognize extracellular proteins may have promotive or suppressive role in cancer cells. Isn't it true?

In Conclusions (line 744 – 748), it may be much better if you find any relationship among issues (i), (ii), and (iii).

Minor comment:

Line 257: 'surviving' may be 'survivin'?

Chen et al. have developed SEREX method and identified NY-ESO-1 for the first time. Following paper should be cited.

Chen YT, Scanlan MJ, Sahin U, Türeci O, Gure AO, Tsang S, et al. A testicular antigen aberrantly expressed in human cancers detected by autologous antibody screening. Proc Natl Acad Sci USA. 1997;94:1914–8. doi: 10.1073/pnas.94.5.1914 PMID: 9050879

Round 2

Reviewer 1 Report

The authors have revised the manuscript to the best of their ability based on the information available about the available accuracy of the of the biomarkers.